# Prevalence and correlates of SARS-CoV-2 seropositivity among people who inject drugs in the San Diego-Tijuana border region

Steffanie A. Strathdee[1]*, Daniela Abramovitz[1], Alicia Harvey-Vera[1,2,3], Carlos F. Vera[1], Gudelia Rangel[3,4], Irina Artamonova[1], Antoine Chaillon[1], Caroline Ignacio[5], Alheli Calderon[1,6], Natasha K. Martin[1], Thomas L. Patterson[7]

1 Division of Infectious Diseases and Global Public Health, Department of Medicine, University of California San Diego, La Jolla, CA, United States of America, 2 Facultad de Medicina, Campus Tijuana, Universidad Xochicalco, Baja California, Mexico, 3 United States-Mexico Border Health Commission, Tijuana, Baja California, Mexico, 4 Departamento de Estudios de Población, El Colegio de la Frontera Norte, Tijuana, Baja California, Mexico, 5 Department of Medicine, San Diego Center for AIDS Research Translational Virology Core, University of California San Diego, La Jolla, CA, United States of America, 6 School of Public Health, San Diego State University, San Diego, California, United States of America, 7 Department of Psychiatry, University of California, San Diego, La Jolla, CA, United States of America

* sstrathdee@health.ucsd.edu

**Data Availability Statement:** All relevant data are within the paper and its Supporting information files. If required, you may list Sharon Park, staff at

## Abstract

### Background

People who inject drugs may be at elevated SARS-CoV-2 risk due to their living conditions and/or exposures when seeking or using drugs. No study to date has reported upon risk factors for SARS-CoV-2 infection among people who inject drugs.

### Methods and findings

Between October, 2020 and June, 2021, participants aged ≥18 years from San Diego, California, USA and Tijuana, Baja California, Mexico who injected drugs within the last month underwent interviews and testing for SARS-CoV-2 RNA and antibodies. Binomial regressions identified correlates of SARS-CoV-2 seropositivity.

### Results

Of 386 participants, SARS-CoV-2 seroprevalence was 36.3% (95% CI: 31.5%-41.1%); 92.1% had detectable IgM antibodies. Only 37.5% had previously been tested. Seroprevalence did not differ by country of residence. None tested RNA-positive. Most (89.5%) reported engaging in ≥1 protective behavior [e.g., facemasks (73.5%), social distancing (46.5%), or increasing handwashing/sanitizers (22.8%)]. In a multivariate model controlling for sex, older age, and Hispanic/Latinx/Mexican ethnicity were independently associated with SARS-CoV-2 seropositivity, as was engaging in sex work (AdjRR: 1.63; 95% CI: 1.18–2.27) and having been incarcerated in the past six months (AdjRR: 1.49; 95% CI: 0.97–2.27). Comorbidities and substance using behaviors were not associated with SARS-CoV-2 seropositivity.

UCSD, as a non-author institutional contact for additional or continued data access in the interest of maintaining long-term data accessibility. You may contact her at shp025@health.ucsd.edu or at 858-246-2622.

**Funding:** SAS, DA, AHV, CFV, GR, IA, AC, AC, NKM, TLP all acknowledge funding from the National Institute on Drug Abuse (R01DA049644-S1). CI and AC acknowledge funding from the National Institute of Allergy and Infectious Disease (P30 AI036214). SAS, DA, CFV, IA, and TLP all acknowledge funding from RADxUP (R01 DA049644-02S2). The funders had no role in study design, data collection, analysis, interpretation, decision to publish, or preparation of the manuscript.

**Competing interests:** The authors have declared that no competing interests exist.

## Conclusions

In this community-based study of people who inject drugs in the San Diego-Tijuana border region, over one third were SARS-CoV-2 seropositive, exceeding estimates from the general population in either city. We found no evidence that substance use behaviors were associated with an elevated risk of SARS-CoV-2 infection, but observed that circumstances in the risk environment, notably sex work and incarceration, were independently associated with higher SARS-CoV-2 seroprevalence. Our findings suggest that a binational policy response to COVID-19 mitigation is warranted beyond the closure of the U.S.-Mexico border. Furthermore, decriminalizing sex work and drug use could reduce the burden of COVID-19 among people who inject drugs.

## Introduction

The disproportionate burden of COVID-19 on under-represented minorities is well documented [1–3]. Substance users may also experience greater COVID-19 related morbidity and mortality. An analysis of data from the U.S. National Survey on Drug Use and Health found that those with opioid and methamphetamine use disorders were more likely to have underlying conditions identified as risk factors for COVID-19 severity and mortality [4]. A recent study found that compared to other patients, people with substance use disorders were more likely to experience breakthrough infections following COVID-19 vaccination, due at least in part to the high prevalence of comorbid conditions [5].

Data are lacking on whether people with substance use disorders are more vulnerable to acquiring and transmitting SARS-CoV-2 due to their living conditions (e.g., homelessness, incarceration) and drug-related behaviors (e.g. smoking, vaping, sharing drug paraphernalia, sex work) [6,7]. In a U.S. nation-wide study, those with substance use disorders, especially African Americans and opioid users, were at increased risk of COVID-19 [8]. However, this study could not determine whether these factors were independently associated with COVID-19 diagnosis since risk factor data were not available, and the sample was restricted to patients accessing health care. In a telephone survey of active and former drug injectors in Baltimore, Maryland, compared to former substance users, those who were actively using substances were less likely to report social distancing, which could increase their risk of acquiring SARS-CoV-2 [7]. In an online survey of adults residing in the Northeastern U.S., daily users of opioids and alcohol were less likely to adhere to COVID-19 related stay-at-home orders, and stimulant users were more likely to report having tested positive for SARS-CoV-2 [9]. There are also reports that the COVID-19 pandemic has interrupted global drug supplies as well as harm reduction and addiction treatment services, which could alter drug use patterns [10,11].

Reports in the literature of SARS-CoV-2 prevalence among people who inject drugs are sparse. In a study of needle exchange clients in Stockholm, Sweden, SARS-CoV-2 prevalence was 5.4% [12], whereas in a survey of people who inject drugs in England, Wales and Northern Ireland, 11% reported that they had tested positive for COVID-19 [13]. U.S. estimates among people who are homeless or living in shelters range from 2%-14% [14–16]. We could not identify any publications that reported risk factors for SARS-CoV-2 among people who inject drugs.

As the eighth largest city in the U.S. and the largest situated on the Mexico-US border, San Diego County, California is home to approximately 1.42 million people. Its counterpart in

Baja California, Mexico is Tijuana, a city of ~1.5 million people, located twenty minutes from San Diego and located on a major drug trafficking route. In an attempt to limit SARS-CoV-2 transmission, the U.S.-Mexico border was closed to essential travel on March 21, 2020 and remained closed for the eight months duration of the study. Modeling of the SARS-CoV-2 epidemic in San Diego County suggests that prevalence was below 5% through July, 2020 then increased to approximately 20% by January, 2021 (California Department of Public Health, unpublished data). In a household study conducted in Tijuana in February 2021, SARS-CoV-2 seroprevalence was 21% [17].

Most epidemiologic research on COVID-19 has been conducted on inpatient samples and has focused on upper-income countries. We studied prevalence and correlates of SARS-CoV-2 infection in a binational community-based study of people who inject drugs residing in San Diego and Tijuana. We hypothesized *a priori* that residents of San Diego who had recently crossed the border to inject drugs in Tijuana and those who were homeless, traded sex or injected drugs more often would be more likely to test SARS-CoV-2 seropositive.

## Materials and methods

### Participants and eligibility

Data collection took place between October 28, 2020 and June 16, 2021 in San Diego and Tijuana by trained interviewers who were residents of either city, using street outreach and mobile vans.

A short screener was used to identify participants who were eligible for study participation. Eligible participants were required to be aged ≥18 or older, report injecting drugs within the last month (as evidenced through injection stigmata) and report living in San Diego County or Tijuana. We sought to enroll participants residing in San Diego who reported having crossed the border to inject drugs in Tijuana within the last two years as well as those from either city who reported not having used illicit drugs on the other side of the border.

### Data collection

After screening was conducted, eligible participants provided written informed consent and underwent interviewer-administered surveys at baseline and approximately one week later to minimize participant burden. Computer assisted personal interviewing was used to minimize socially desirable responding. Interviews were conducted in person, with plexiglass dividers separating the interviewer and participant, who were both required to wear facemasks.

### Survey measures

Survey items were developed based on a previous study [18], as well as the C3PNO study [19] (refer to Supplemental Materials). Measures included sociodemographics, potential COVID-19 exposures (e.g., injection and non-injection drug use, sharing injection equipment, incarceration, sex work), vulnerabilities (e.g., homelessness, cross-border, mobility, food insecurity [8,20], impact of COVID-19 on income and housing), and protective behaviors (e.g., social distancing, use of facemasks and handwashing/sanitizers). Before beginning data collection, the survey was translated into Spanish, back-translated into English, and verified for accuracy by bilingual team members. To eliminate duplicate enrollments, identifying information was collected from participants as well as photographs, which were stored separately from survey data in encrypted, password protected files. Participants were compensated $20 USD and were provided with photo ID cards embossed with the study logo and contact information. Protocols were approved by the Human Research Protections Program and Biosafety Committee at

the University of California San Diego (UCSD) and the institutional review board at Xochi-calco University.

**Biological measures.** Venous blood samples were obtained by trained phlebotomists to conduct serology for SARS-CoV-2, HIV and HCV. Pre- and post-test counseling was provided following national guidelines in the U.S. and Mexico. Rapid testing for HIV and HCV was done during the study visit, enabling participants to immediately obtain their test results. Biological samples were batched and stored at -80 degrees Celsius and shipped weekly on dry ice for SARS-CoV-2 RNA and antibody detection.

## HIV and HCV serology

Rapid HIV and HCV tests were conducted by trained study staff using the Miriad® HIV/HCV Antibody InTec Rapid Anti-HCV Test (Avantor, Radnor, PA). Reactive and indeterminate tests underwent a second rapid test with Oraquick® HIV or Oraquick® HCV, respectively (Orasure, Bethlehem, PA). HIV and HCV rapid test reactive were sent for confirmatory testing at UCSD's Center for AIDS Research (CFAR) laboratory.

## SARS-CoV-2 RNA detection

Participants were instructed on how to self-collect anterior nasal swabs in the presence of study staff. Swabs which were placed in 3mL of viral transport media for temporary storage, before being shipped for testing at the UCSD CFAR laboratory. RT-PCR was conducted using a pooling approach based on the Fluxergy system® (Irvine, CA) to detect SARS-CoV-2 RNA.

## SARS-CoV-2 antibody detection

Serology was conducted by Genalyte® (San Diego, CA), using their Maverick™ Multi-Antigen Serology Panel [21] that detects IgG and IgM antibodies to five SARS-CoV-2 antigens (Nucleocapsid, Spike S1-S2, Spike S1, Spike S1-RBD, Spike S2) within a multiplex format based on photonic ring resonance. A machine learning algorithm was used to call results using the Random Forest Ensemble method with 3000 decision trees [22].

## Statistical analysis

SARS-CoV-2 prevalence was calculated with 95% confidence intervals (CIs) based on the Binomial distribution. Those testing indeterminate were excluded. The Cochran-Armitage test was used to assess trends in SARS-CoV-2 seroprevalence.

Characteristics of participants testing SARS-CoV-2 seropositive versus seronegative were compared using Wilcoxon Rank Sum for continuous variables and Chi-square or Fisher's Exact tests for categorical variables. Univariate and multivariable binomial regressions with robust standard error estimation via generalized estimating equations were performed to identify factors associated with SARS-CoV-2 seropositivity. Variables attaining <10% significance were considered for inclusion in multivariable models. All potential interactions between variables in the final model were assessed. Multi-collinearity was assessed using diagnostics such as largest condition index and variance inflation factors. Although site-specific models were examined, results were presented for the overall sample since associations were generally similar. All analyses were conducted using SAS version 9.4.

## Role of funding source

The funders had no involvement in the study design, collection, analysis, interpretation or writing of this report, nor the decision to submit the paper for publication.

## Results

### Biologic testing

Of 405 participants tested, none had detectable SARS-CoV-2 RNA. Considering serologic evidence of SARS-CoV-2 infection, 19 (4.7%) tested indeterminate and were excluded from further analysis. These participants did not differ significantly from those who were included, with the exception that those testing indeterminate were less likely to inject heroin compared to the remainder of participants (68.4% vs. 87.4%, P = 0.04). Detectable SARS-CoV-2 IgG and/or IgM antibodies were observed in 36.3% [95% confidence interval (95% CI): 31.5%-41.1%], of whom the majority had IgM (92.1%). Of the 140 testing SARS-CoV-2 seropositive, only 37.5% had previously been tested, and 26.8% reported ≥1 current symptom consistent with COVID-19. There was no significant trend in seroprevalence over time (p = 0.80).

### Descriptive statistics

Of the 386 subjects included in the analysis, 63.5% lived in San Diego County. Most were male (74.1%) and Hispanic/Latinx/Mexican (71.5%). Median age was 43 (inter-quartile range [IQR: 35–51]. Over half reported that COVID-19 had adversely affected their housing or income. During the last six months, 39.1% were homeless, 8.3% were incarcerated and 12.4% traded sex (Table 1). Of the San Diego residents, 50.8% reported having crossed the border to inject drugs in Tijuana within the last six months.

Although 89.5% reported engaging in ≥1 protective behavior [e.g., face masks (73.5%)], only 46.5% reported social distancing, 22.8% increased handwashing/hand sanitizers, and 9.0% reported avoiding sharing drug paraphernalia in the prior six months (Table 1).

**Bivariate analyses.** Older age and identifying as Hispanic, Latinx, or Mexican were significantly associated with SARS-CoV-2 seropositivity (Table 1). Considering potential community exposures in the past six months, those engaging in sex work were significantly more likely to test seropositive compared to those who did not (17.9% vs. 9.3%, p = 0.01, [Relative Risk (RR): 1.53; 95% CI: 1.12–2.09]. Being incarcerated was associated with marginally higher SARS-CoV-2 seropositivity (RR: 1.43; 95% CI: 0.99–2.09). We did not observe injection drug use, smoking, vaping specific drugs or any other substance use behaviors to be associated with seroprevalence, nor were any co-morbidities (i.e., HIV, HCV, Type 2 diabetes, hypertension). Protective behaviors were not significantly associated with SARS-CoV-2 serostatus, with the exception that those testing SARS-CoV-2 seropositive were more likely to report having stopped smoking in the last six months and to have had a COVID-19 test.

**Multivariate analysis.** In a multivariate model that controlled for sex (Table 2), each year in age was associated with a 2% increase in SARS-CoV-2 seropositivity (Adjusted relative risk (AdjRR): 1.02; 95% CI: 1.01–1.03). Identifying as Hispanic/Latinx/Mexican was also independently associated with SARS-CoV-2 seropositivity (AdjRR: 1.53; 95% CI: 1.09–2.15), as was engaging in sex work in the past six months (adjRR: 1.63; 95% CI: 1.18–2.27). Being incarcerated in the last six months remained marginally significant (AdjRR: 1.49; 95% CI: 0.97–2.27). When country of residence was forced into the model, all significant associations held except ethnicity. Excluding the nine participants who reported having received at least one COVID-19 vaccine did not appreciably change parameter estimates.

Given that sex work was independently associated with SARS-CoV-2 seropositivity, we examined this subgroup more closely. Compared to non-sex workers, sex workers were just as less likely report protective behaviors but were significantly more likely to report being exposed to someone with COVID-19 (20% vs. 3.8%, p = 0.001) or to have low/very low food security (94.3% vs. 77.9% p = 0.02).

**Table 1. Characteristics associated with SARS-CoV-2 Sero-positivity among people who inject drugs in San Diego, California and Tijuana, Mexico.**

| Baseline Characteristics | SARS-CoV-2 Seropositive N = 140 | SARS-CoV-2 Seronegative N = 246 | Total N = 386 | Univariate RR (95% CI) |
|---|---|---|---|---|
| *Sociodemographics* | | | | |
| Male | 104(74.3%) | 182(74.0%) | 286(74.1%) | 1.01 (0.75,1.37) |
| Median Age (IQR)[P] | 45.0(37.0,53.0) | 42.0(34.0,50.0) | 43.0(35.0,51.0) | 1.02 (1.00,1.03)[¥] |
| Hispanic/Latinx/Mexican[P] | 111(79.3%) | 165(67.1%) | 276(71.5%) | 1.53 (1.08,2.15) |
| Speaks English | 98(70.0%) | 185(75.2%) | 283(73.3%) | 0.85 (0.64,1.13) |
| Born in the US | 61(43.6%) | 129(52.4%) | 190(49.2%) | 0.80 (0.61,1.04) |
| Primary residence in San Diego | 88(62.9%) | 157(63.8%) | 245(63.5%) | 1.00 (0.96,1.04) |
| Highest year of school completed (IQR) | 11.0(8.0,12.0) | 11.0(7.0,12.0) | 11.0(7.0,12.0) | 0.92 (0.65,1.30) |
| Married or common law | 25(17.9%) | 49(19.9%) | 74(19.2%) | 1.02 (0.78,1.33) |
| Average monthly income <500 USD | 75(53.6%) | 130(52.8%) | 205(53.1%) | 0.97 (0.74,1.28) |
| Median years lived in Study Location (IQR) | 30.0(10.0,45.0) | 26.5(10.0,40.0) | 28.0(10.0,41.0) | 1.01 (1.00,1.01)[¥] |
| *Potential Exposures* | | | | |
| Homeless* | 50(35.7%) | 101(41.1%) | 151(39.1%) | 0.86 (0.65,1.14) |
| Incarcerated*[P] | 16(11.5%) | 16(6.5%) | 32(8.3%) | 1.43 (0.99,2.09) |
| Median # of people in household (IQR)* | 2.0(1.0, 4.0) | 2.0(1.0, 5.0) | 2.0(1.0, 4.0) | 1.00 (1.00,1.00)[¥] |
| Low/very low food security | 88(78.6%) | 171(80.3%) | 259(79.7%) | 0.93 (0.65,1.34) |
| Engaged in sex work*[P] | 25(17.9%) | 23(9.3%) | 48(12.4%) | 1.53 (1.12,2.09) |
| Client of sex worker* | 8(5.7%) | 10(4.1%) | 18(4.7%) | 1.24 (0.73,2.11) |
| Exposed to someone with COVID-19 | 6(5.4%) | 12(5.6%) | 18(5.5%) | 0.97 (0.49,1.89) |
| *Impact of Pandemic* | | | | |
| Housing situation worse | 91(65.0%) | 140(56.9%) | 231(59.8%) | 1.25 (0.94,1.65) |
| Income worse | 98(70.5%) | 149(62.3%) | 247(65.3%) | 1.27 (0.94,1.71) |
| *Substance Use* | | | | |
| Smokes cigarettes | 118(84.3%) | 222(90.2%) | 340(88.1%) | 0.73 (0.52,1.01) |
| Smoked or vaped marijuana* | 67(47.9%) | 139(56.5%) | 206(53.4%) | 0.80 (0.62,1.04) |
| Smoked/snorted/inhaled heroin*[P] | 32(22.9%) | 73(29.7%) | 105(27.2%) | 0.79 (0.57,1.10) |
| Smoked/snorted/inhaled/vaped meth* | 86(61.4%) | 150(61.0%) | 236(61.1%) | 1.01 (0.77,1.33) |
| Smoked/snorted/inhaled crack/cocaine* | 15(10.7%) | 22(8.9%) | 37(9.6%) | 1.13 (0.75,1.71) |
| Injected heroin* | 123(87.9%) | 222(90.2%) | 345(89.4%) | 0.86 (0.58,1.27) |
| Injected fentanyl* | 27(19.3%) | 50(20.3%) | 77(19.9%) | 0.96 (0.68,1.34) |
| Median age at first injection | 20.0(17.0,27.0) | 19.0(17.0,25.0) | 20.0(17.0,26.0) | 1.01 (0.99,1.02) |
| #Times injected drugs per day | 2.5(0.3, 4.0) | 2.5(0.7, 4.0) | 2.5(0.3, 4.0) | 0.94 (0.87,1.02)[¥] |
| Visited shooting galleries* | 12(8.6%) | 19(7.7%) | 31(8.0%) | 1.07 (0.67,1.71) |
| Used hit doctor* | 25(17.9%) | 46(18.8%) | 71(18.4%) | 0.96 (0.68,1.36) |
| Crossed border to inject drugs* | 60(42.9%) | 97(39.4%) | 157(40.7%) | 1.09 (0.84,1.43) |
| *Co-Morbidities* | | | | |
| HIV-antibody positive | 15(10.7%) | 17(6.9%) | 32(8.3%) | 1.33 (0.89,1.97) |
| HCV-antibody positive | 47(33.6%) | 86(35.1%) | 133(34.5%) | 0.96 (0.72,1.27) |
| Diabetes[Y] | 5(4.5%) | 10(4.7%) | 15(4.6%) | 0.97 (0.46,2.01) |
| Asthma[Y] | 8(7.1%) | 22(10.3%) | 30(9.2%) | 0.76 (0.41,1.40) |
| Hypertension[Y] | 13(11.6%) | 25(11.7%) | 38(11.7%) | 0.99 (0.62,1.58) |
| *Preventive Measures* | | | | |
| Practiced social distancing[Y] | 52(46.4%) | 99(46.5%) | 151(46.5%) | 1.00 (0.74,1.35) |
| Wore face mask[Y] | 85(75.9%) | 154(72.3%) | 239(73.5%) | 1.13 (0.79,1.62) |
| Increased handwashing/sanitizer[Y] | 24(21.4%) | 50(23.5%) | 74(22.8%) | 0.93 (0.64,1.34) |
| Stocked up on drugs[Y] | 18(16.1%) | 35(16.5%) | 53(16.4%) | 0.98 (0.65,1.47) |
| Stocked up on harm reduction supplies[Y] | 24(21.4%) | 38(17.9%) | 62(19.1%) | 1.15 (0.81,1.65) |

*(Continued)*

**Table 1.** (Continued)

| Baseline Characteristics | SARS-CoV-2 Seropositive N = 140 | SARS-CoV-2 Seronegative N = 246 | Total N = 386 | Univariate RR (95% CI) |
|---|---|---|---|---|
| Stopped smoking (current smokers)[YP] | 6(6.3%) | 4(2.1%) | 10(3.5%) | 1.86 (1.09,3.17) |
| Avoided sharing drug paraphernalia[Y] | 11(9.8%) | 18(8.5%) | 29(9.0%) | 1.11 (0.68,1.81) |
| Engaged in ≥1 protective behavior[Y] | 101(90.2%) | 190(89.2%) | 291(89.5%) | 1.07 (0.64,1.79) |
| Had a prior COVID-19 test[YP] | 42(37.5%) | 63(29.6%) | 105(32.3%) | 1.26 (0.93,1.70) |

[*]past 6 months

[Y]Missing values n = 62

[¥]Per year increase

[P]P-value<0.10.

## Discussion

In this community-based study of people who inject drugs in the San Diego-Tijuana border region, over one third had detectable SARS-CoV-2 antibodies, which exceeds estimates from the general population in either city [17]. This suggests that a binational policy response to COVID-19 is warranted beyond the closure of the U.S.-Mexico border. We did not observe substance use behaviors, such as smoking, vaping, or use of specific drugs such as opiates or stimulants to be associated with an elevated risk of SARS-CoV-2 infection, as others have hypothesized [9]. Instead, we observed that circumstances in the risk environment, notably sex work and incarceration, were independently associated with higher SARS-CoV-2 seroprevalence.

To our knowledge, this is the first study to show that sex work is independently associated with higher SARS-CoV-2 seroprevalence after controlling for potential confounders such as sex, age, and ethnicity. In an unpublished study in Denmark, SARS-CoV-2 seroprevalence was higher among sex workers (12.2%) than people experiencing homelessness (6.8%) or the general population (2.9%) [23], but the independent effects of these and other factors were not assessed.

Although the majority of our sample reported that COVID-19 had greatly affected their income and housing, sex workers may have faced additional hardships following the closure of the U.S.-Mexico border because of their reliance on sex tourism, and since social distancing is not possible during sexual transactions. Compared to the rest of the sample, significantly higher proportions of sex workers reported low or very low food security, and they were more likely to report having been exposed to someone with COVID-19 than others. This suggests

**Table 2. Factors Independently associated with SARS-CoV-2 Seropositivity among people who inject drugs in San Diego, CA and Tijuana, Mexico.**

| Baseline Characteristics | Adjusted RR[**] (95% CI) |
|---|---|
| Male | 1.02 (0.76, 1.37) |
| Age[¥] | 1.02 (1.01, 1.03) |
| Hispanic/Latinx/Mexican | 1.53 (1.09, 2.15) |
| Engaged in sex work[*] | 1.63 (1.18, 2.27) |
| Incarcerated[*] | 1.49 (0.97, 2.27) |

[*]past 6 months

[**]variables in the multivariable model were adjusted for all the variables in the model.

[¥] Per year increase.

that sex workers may have engaged in higher risk behaviors to support themselves, placing them at greater risk of SARS-CoV-2 infection. In a study of female sex workers in Kenya, sexual transactions declined during the pandemic [24] and those most reliant on sex work reported greater food insecurity, which is consistent with our findings. Among female sex workers in Nigeria, those who had knowledge about COVID-19 were significantly more likely to wear face masks, but less than half did so [25].

Despite concerns about sex workers' vulnerability to COVID-19, few countries provide them aid [24,26]. In Thailand, the Netherlands and Japan, sex workers were included in COVID-19 government-sponsored support programs [26], but they are excluded in countries where sex work is criminalized, such as the U.S. Although sex work is quasi-legal in Tijuana's red light district, those using drugs are less likely to obtain sex work permits and instead work outside of commercial establishments that could offer protections [27]. In an earlier study, we found that female sex workers in Tijuana who inject drugs were more vulnerable to offers of unprotected sex in exchange for more money or drugs compared to those that did not inject [28].

Our finding that recent incarceration was independently associated with SARS-CoV-2 seropositivity could reflect institutional exposures. Over-crowding was implicated in COVID-19 outbreaks in California correctional institutions [29], including those in the U.S.-Mexico border region. Mitigation included early release, halting intakes, and eliminating bail, but a recent analysis suggests that considerable potential for SARS-CoV-2 exposures persists, at least in the California prison system [29]. Similar measures have not been undertaken in Baja California. Taken together, our findings suggest that decriminalization of drug use and sex work and increasing their access to social protection programs could reduce SARS-CoV-2 risk among people who inject drugs and sex workers, as others have proposed [26,30].

SARS-CoV-2 seroprevalence did not differ by country of residence in our study, perhaps reflecting close social ties between San Diego and Tijuana communities despite the closure of the U.S.-Mexico border. In a phylogenetic analysis of SARS-CoV-2 sequences from the U.S. and Mexico, those obtained from the Mexican state of Baja California were more closely related to San Diego than to Mexico City [31]. Although one half of the San Diego residents in our study had crossed the border and injected drugs in Tijuana in the last six months, we found no evidence that cross-border mobility was associated with SARS-CoV-2 seropositivity.

Consistent with other studies, subjects who were older [8] and who identified as Hispanic, Latinx, or Mexican [2] were significantly more likely to test SARS-CoV-2 seropositive. This demonstrates that even among the lowest socioeconomic strata in the US and Mexico, ethnic disparities in SARS-CoV-2 seroprevalence persist.

Our study also found that over half of those testing SARS-CoV-2 seropositive had not been tested prior to study enrolment. These findings underscore the need to improve community outreach to provide testing, vaccines, and treatment, for example using mobile syringe exchange programs.

## Limitations

Our ability to detect some associations was limited due to statistical power. Ours was a non-random sample and the cross-sectional study design precludes drawing causal inferences. Participants experiencing symptoms may have changed behaviors, such as stopped smoking or sought COVID-19 testing. Some misclassification may have also occurred since behaviors such as current and past substance use were self-reported.

Of note, none of the self-collected swabs tested positive for SARS-CoV-2 RNA. The sensitivity of the pooling approach could have been impacted by: 1) the viral load of any particular

infected individual; 2) the consistency with which swabs were obtained; 3) storage, shipping, and transport conditions; or 4) of diluting out (via pooling) any viral SARS-CoV-2 RNA collected below the limit of detection. In this study, we limited our pools to ≤10 swabs, and previously validated our approach with up to 30 samples per pool where the limit of detection was estimated at 2.4 copies/μL [32]. Therefore, it is unlikely that we experienced loss of sensitivity due to pooling.

Since SARS-CoV-2 antibodies may wane over time especially among patients who are asymptomatic or mildly symptomatic [33,34], infections that occurred earlier in the epidemic may have gone undetected. However, the majority of participants testing SARS-CoV-2 seropositive had detectable IgM antibody titers, which is suggestive of recent infection [35]. Some misclassification could have occurred among those testing indeterminate who were recently infected, but these were few in number and would have tended to underestimate SARS-CoV-2 prevalence, dampening any observed associations.

Since COVID-19 vaccines did not become available to adults 18–65 years of age in San Diego County until May 15, 2021 and were not available to most adults in Tijuana until after the study period ended, we were unable to examine the impact of vaccination on SARS-CoV-2 infection. Nevertheless, among the 75 San Diego residents in our sample who were interviewed after May 15, only 9 (9.3%) reported having received at least one COVID-19 vaccine dose. In comparison, the proportion of adults aged ≥18 years who had received at least one COVID-19 vaccine dose in San Diego County through June 24, 2021 exceeded 50% [36].

## Conclusions

We found that most people who inject drugs in the San Diego-Tijuana border region engaged in preventive measures to avoid SARS-CoV-2, but since over one-third had evidence of infection, a binational policy response is warranted beyond the closure of the U.S.-Mexico border. Importantly, social and structural factors in the risk environment were independently associated with SARS-CoV-2, and substance use behaviors were not, suggesting that structural interventions such as decriminalizing sex work and drug use and increasing their access to social protection programs could reduce vulnerability to SARS-CoV-2 among people who inject drugs. Since the proportion of participants who had previously been tested for SARS-CoV-2 or had received COVID-19 vaccine was very low, efforts are needed to mitigate risks and provide COVID-19 testing and vaccines this vulnerable population.

## Supporting information

**S1 File. La Frontera protocol.**
(DOCX)

**S2 File. La Frontera baseline and supplemental surveys.**
(PDF)

**S3 File. La Frontera data set.**
(SAS7BDAT)

## Acknowledgments

The authors gratefully acknowledge the La Frontera study team and participants in San Diego and Tijuana, staff at Genalyte and Fluxergy for assistance interpreting laboratory results, laboratory staff at the Center for AIDS Research, Dr Pamina Gorbach for providing some COVID

survey measures from the C3PNO COVID-19 Survey (available at https://tools.niehs.nih.gov/dr2/index.cfm/resource/22690), Dr. Davey Smith for helpful suggestions on the study design and Sharon Park for assistance with manuscript preparation.

## Author Contributions

**Conceptualization:** Steffanie A. Strathdee, Gudelia Rangel, Natasha K. Martin, Thomas L. Patterson.

**Data curation:** Daniela Abramovitz, Alicia Harvey-Vera, Carlos F. Vera, Gudelia Rangel, Irina Artamonova.

**Formal analysis:** Daniela Abramovitz, Natasha K. Martin.

**Funding acquisition:** Steffanie A. Strathdee.

**Investigation:** Caroline Ignacio, Alheli Calderon, Natasha K. Martin.

**Methodology:** Steffanie A. Strathdee, Alicia Harvey-Vera, Gudelia Rangel, Antoine Chaillon, Caroline Ignacio, Alheli Calderon, Natasha K. Martin, Thomas L. Patterson.

**Project administration:** Steffanie A. Strathdee.

**Resources:** Caroline Ignacio.

**Software:** Daniela Abramovitz, Irina Artamonova.

**Supervision:** Steffanie A. Strathdee.

**Validation:** Daniela Abramovitz, Irina Artamonova, Natasha K. Martin.

**Writing – original draft:** Steffanie A. Strathdee.

**Writing – review & editing:** Steffanie A. Strathdee, Daniela Abramovitz, Alicia Harvey-Vera, Carlos F. Vera, Gudelia Rangel, Irina Artamonova, Antoine Chaillon, Caroline Ignacio, Alheli Calderon, Natasha K. Martin, Thomas L. Patterson.

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
