## [Decision Letter · Decision Letter 0]

15 Oct 2021

PONE-D-21-25385Prevalence and Correlates of SARS-CoV-2 Seropositivity among People who Inject Drugs in the San Diego-Tijuana Border RegionPLOS ONE

Dear Dr. Strathdee,

Thank you for submitting your manuscript to PLOS ONE. After careful consideration, we feel that it has merit but does not fully meet PLOS ONE’s publication criteria as it currently stands. Therefore, we invite you to submit a revised version of the manuscript that addresses the points raised during the review process.

Additional Editor Comments (if provided):

Reviewers indicate several areas that can strengthen this manuscript. Although they differ in their suggestions regarding - minor and major revision, I do think that the comments from both are helpful and appropriate and can be easily addressed. I do disagree that the 'a priori' note needs to be removed. I think this is a good manuscript and hope the authors will revise and resubmit. 

Please submit your revised manuscript by Nov 29 2021 11:59PM If you will need more time than this to complete your revisions, please reply to this message or contact the journal office at plosone@plos.org. Please include the following items when submitting your revised manuscript:A rebuttal letter that responds to each point raised by the academic editor and reviewer(s). You should upload this letter as a separate file labeled 'Response to Reviewers'.A marked-up copy of your manuscript that highlights changes made to the original version. You should upload this as a separate file labeled 'Revised Manuscript with Track Changes'.An unmarked version of your revised paper without tracked changes. You should upload this as a separate file labeled 'Manuscript'.

We look forward to receiving your revised manuscript.

Kind regards,

Kimberly Page, PhD, MPH

Academic Editor

PLOS ONE

Journal Requirements:

Furthermore, please provide additional information regarding the steps taken to validate the questionnaire.

4. We note that you have referenced (California Department of Public Health, unpublished data which has currently not yet been accepted for publication. Please remove this from your References and amend this to state in the body of your manuscript: (ie “California Department of Public Health [Unpublished]”) as detailed online in our guide for authors

Reviewers' comments:

Reviewer's Responses to Questions

**Comments to the Author**

1. Is the manuscript technically sound, and do the data support the conclusions?

Reviewer #1: Yes

Reviewer #2: No

2. Has the statistical analysis been performed appropriately and rigorously? 

Reviewer #1: Yes

Reviewer #2: Yes

3. Have the authors made all data underlying the findings in their manuscript fully available?

Reviewer #1: No

Reviewer #2: Yes

4. Is the manuscript presented in an intelligible fashion and written in standard English?

Reviewer #1: Yes

Reviewer #2: Yes

5. Review Comments to the Author

Reviewer #1: The manuscript makes an important contribution to the COVID literature among the PWID population with high rates of sex work. One important finding is that injection drug use or other substance use behaviors were not associated with seroprevalence; rather circumstances in the risk environment, notably sex work and incarceration, were associated with significantly higher SARS-CoV-2 seroprevalence.

Several clarification and suggestions are below:

• Clarification on recruitment: The authors indicated that street-based recruiting was used between October 28, 2020 and June 16, 2021 (a period when the border was closed to essential travel). It’s unclear how recruitment was done for participants in Tijuana and “those from either city who reported not having used illicit drugs on the other side of the border”. Was all recruitment done in San Diego?

• Clarification needed on “personal interviewing” – were individuals interviewed in person during this period? Or via Zoom?

• Did the study collect identifiers in order to ascertain duplicate enrollments?

• In the limitations section, the authors reported the majority of participants had IgM; however, this is not reported in the results. Given that there were no active infections reported in the sample, it would be interesting to report how many (if any) were individuals had only IgM and not IgG as an estimate of recent infection.

• Minor: The recruitment period includes a few months where individuals may have gotten vaccinations. While most antibody tests were not designed to specifically detect antibodies as a result of vaccination, is the possible that particular antibody tests used may have yielded a false positive? This would be easily seen if the 9 individuals who reported if the participant were all positive relative to the rest of the sample. In the limitations section, the authors reported on the vaccinated portions of the sample, but not in the results. Nonetheless, the authors did report that excluding nine participants who reported having received at least one COVID-19 vaccine did not appreciably change parameter estimates in multivariable analysis.

• Minor Suggestion: Table 1 reports column %, which provide snapshot of the sample distribution. It would be interesting to more directly compare sub-groups (row%). One suggestion would be just to the univariate RR from Table 2 to Table 1.

Reviewer #2: Thank you for giving me the opportunity to review the manuscript titled “Prevalence and Correlates of SARS-CoV-2 Seropositivity among People who Inject Drugs in the San Diego- Tijuana Border Region.” This manuscript presents relevant information related to the prevalence and correlates of SARS-CoV-2 in PWID, a vulnerable population at increased risk for infection and increased morbidity and mortality. This study is innovative in terms that it provides novel information related to PWID living in an understudied region and at an increased vulnerability. However, there are some issues in this paper that limit my enthusiasm.

Introduction

Overall, this section is extremely short and does not provide the background necessary to understand why this research study exploring the prevalence and correlates of SARS-CoV-2 among people who inject drugs is necessary.

I recommend including estimates and correlates of SARS-CoV-2 obtained from earlier studies involving people with substance use disorder. I would also suggest including the limitations in earlier research studies (e.g., mostly conducted in developed countries), which would strengthen the Introduction section.

The Introduction section should highlight the fact that SARS-CoV-2 could be a serious threat to health among those people who use drugs.

There is not a lack of data on the effects of SARS-CoV-2 on people with substance use disorders. In fact, there is a growing amount of research focusing on this topic. It would be important to reframe the second sentence and specify in which specific area lacks data.

Methods

There are several aspects related to the study procedures that are ambiguous. The order of the assessments is unclear. It would be very helpful to include a paragraph describing the order of the assessments.

The inclusion criteria are unclear for participating in this study. It is presented that individuals who recently injected any drugs (within 30 days) are eligible, but also those who injected drugs in the last two years as well as those who reported not having used illicit drugs. I would recommend listing the inclusion and exclusion criteria to improve readability.

Did the participants provide written informed consent?

It would be important to cite where the questionnaire came from.

Participants self-collected nasal swabs. Were these participants trained in the use of the swabs to ensure they were collecting the biological samples correctly?

There is no description of when and by whom the SARS-CoV-2 antibody, HIV, and HCV serology tests were conducted.

Data analyses

It would be interesting to explore whether the characteristics of the participants with inconclusive SARS-CoV-2 results differ from those with positive results.

Results

I suggest some editing in this section to improve readability. It would be helpful to focus one paragraph only describing the overall sample, then another presenting the bivariate analyses, and a final one describing the multivariate analyses.

Discussion

This section provides information that would have been useful in the Introduction section to provide a background on the topic. This includes the prevalence and correlates of SARS-CoV-2.

This section presents 7 different findings obtained from the current study. It is unclear what the take home message is. I would suggest rewording and reorganizing this section to focus on the most relevant findings.

It should be noted that the reasons for these 7 different findings are not fully described.

Recent or past drug use is self-reported. This should be included as a study limitation.

Conclusions

The conclusions section does not fully mirror what is presented in the Discussion section and reads as a different paper. This last paragraph should focus on the findings related to the prevalence of SARS-CoV-2 and its correlates among PWID. The additional information is distracting.

Minor comments:

- Remove “a priori” from the sentence “We hypothesized a priori that…”

- Include the software used to conduct the data analyses

6. PLOS authors have the option to publish the peer review history of their article (what does this mean?). If published, this will include your full peer review and any attached files.

Reviewer #1: No

Reviewer #2: **Yes: **Irene Pericot-Valverde

---

## [Author Response · Author response to Decision Letter 0]

27 Oct 2021

Editorial Revisions

Response: We have adhered to the journal’s requirements. 

Furthermore, please provide additional information regarding the steps taken to validate the questionnaire.

Response: We have uploaded English and Spanish versions of the baseline and supplemental survey as supporting information, and have indicated this in the text. Please note that the survey was formatted so it could be programmed into QDS (CAPI software) and the format of the surveys we are providing are not intended to be administered with pencil and paper. 

The sociodemographic and behavioral components of the survey was based on previous items used in an earlier study (Proyecto El Cuete). Some of the COVID-19 questions were obtained from the CP30 study (c3pno-covid-19-survey-measures (2).pdf ). The Spanish version of the survey was back-translated into English and reviewed for accuracy. We have added these details to the text and provided citations for validated scales that were assessed in this study (e.g., food insecurity). 

Response: We have made corrections so these statements now match.

4. We note that you have referenced (California Department of Public Health, unpublished data which has currently not yet been accepted for publication. Please remove this from your References and amend this to state in the body of your manuscript: (ie “California Department of Public Health [Unpublished]”) as detailed online in our guide for authors

http://journals.plos.org/plosone/s/submission-guidelines#loc-reference-style.

Response: We have removed this source from the reference list as requested.

Response: No citations were retracted. The reference list is complete.

Review Comments to the Author

Reviewer #1: The manuscript makes an important contribution to the COVID literature among the PWID population with high rates of sex work. One important finding is that injection drug use or other substance use behaviors were not associated with seroprevalence; rather circumstances in the risk environment, notably sex work and incarceration, were associated with significantly higher SARS-CoV-2 seroprevalence.

Several clarification and suggestions are below:

1. Clarification on recruitment: The authors indicated that street-based recruiting was used between October 28, 2020 and June 16, 2021 (a period when the border was closed to essential travel). It’s unclear how recruitment was done for participants in Tijuana and “those from either city who reported not having used illicit drugs on the other side of the border”. Was all recruitment done in San Diego?

Response: The closure of the border did not interrupt data collection. Recruitment was done in both San Diego and Tijuana using staff who were residents of the respective cities. We have clarified this in the text.

2. Clarification needed on “personal interviewing” – were individuals interviewed in person during this period? Or via Zoom?

Response: Interviews were conducted in person. We added the following sentence to clarify:

“Interviews were conducted in person, with plexiglass dividers separating the interviewer and participant, who were both required to wear facemasks.”

3. Did the study collect identifiers in order to ascertain duplicate enrollments?

Response: Identifying information was collected as well as photographs of participants to eliminate duplicate enrolments. We added this text to the Methods. 

4. In the limitations section, the authors reported the majority of participants had IgM; however, this is not reported in the results. Given that there were no active infections reported in the sample, it would be interesting to report how many (if any) were individuals had only IgM and not IgG as an estimate of recent infection.

Response: Unfortunately, Genalyte does not separate IgM and IgG serology in their reports, so we could make this information available. We report in the text that that 92% had detectable IgM antibodies. 

5. Minor: The recruitment period includes a few months where individuals may have gotten vaccinations. While most antibody tests were not designed to specifically detect antibodies as a result of vaccination, is it possible that particular antibody tests used may have yielded a false positive? 

Response: We consulted with the Chief Scientific Officer from Genalyte who shared that the positive predictive value for their antibody assay is 97.2%.

6. Minor Suggestion: Table 1 reports column %, which provide snapshot of the sample distribution. It would be interesting to more directly compare sub-groups (row%). One suggestion would be just to the univariate RR from Table 2 to Table 1.

Response: We have modified the tables accordingly and collapsed Tables 1 and 2.

Reviewer #2: 

7. Introduction

Overall, this section is extremely short and does not provide the background necessary to understand why this research study exploring the prevalence and correlates of SARS-CoV-2 among people who inject drugs is necessary. I recommend including estimates and correlates of SARS-CoV-2 obtained from earlier studies involving people with substance use disorder. 

Response: We appreciated the opportunity to expand the Introduction. As suggested, we have moved text from the Discussion reporting prevalence of SARS-CoV-2 among substance users to the Introduction. We also updated the literature review to cover research published during the three month period since our paper was prepared for submission. The introduction is now more comprehensive.

8. I would also suggest including the limitations in earlier research studies (e.g., mostly conducted in developed countries), which would strengthen the Introduction section.

Response: Thank you. We agree and added this as a limitation of the earlier studies in the Introduction.

9. The Introduction section should highlight the fact that SARS-CoV-2 could be a serious threat to health among those people who use drugs.

Response: We agree and have added this sentence to the Introduction with new accompanying references.

10. There is not a lack of data on the effects of SARS-CoV-2 on people with substance use disorders. In fact, there is a growing amount of research focusing on this topic. It would be important to reframe the second sentence and specify in which specific area lacks data.

Response. At the time our paper was submitted (early August 2021), there were few papers published on the effects of SARS-CoV-2 on people with substance use disorders. We repeated a literature review in October 2021 and revised our Introduction accordingly to include new publications that were relevant. We amended the sentence about how data are lacking as follows: 

“Data is lacking about whether people with substance use disorders are more vulnerable to acquiring and transmitting SARS-CoV-2 due to their living conditions (e.g., homelessness, incarceration) and drug-related behaviors (e.g. smoking, vaping, sharing drug paraphernalia, sex work).”

Our literature review could still not identify any additional papers on risk factors for SARS-CoV-2 among persons with substance use disorders, with the exception of one paper that found that stimulant users had a higher SARS-CoV-2 prevalence. We have now clarified that this is a gap in the literature that our study attempts to fill. 

11. Methods

The order of the assessments is unclear. It would be very helpful to include a paragraph describing the order of the assessments.

Response: We have clarified the order of assessments in the Methods section.

12. The inclusion criteria are unclear for participating in this study. It is presented that individuals who recently injected any drugs (within 30 days) are eligible, but also those who injected drugs in the last two years as well as those who reported not having used illicit drugs. I would recommend listing the inclusion and exclusion criteria to improve readability.

Response: We apologize for the confusion and have revised the section on eligibility to make it clearer. 

13. Did the participants provide written informed consent?

Response: Yes. We have revised the sentence on informed consent to clarify that written consent was obtained. 

14. It would be important to cite where the questionnaire came from.

Response: See response to #2.

15. Participants self-collected nasal swabs. Were these participants trained in the use of the swabs to ensure they were collecting the biological samples correctly?

Response: Yes. We have clarified that study participants were trained about how to collect anterior nasal swabs, which they collected in the presence of study staff. 

16. There is no description of when and by whom the SARS-CoV-2 antibody, HIV, and HCV serology tests were conducted.

Response: We have clarified the timing of the tests, and indicated which were performed by study staff (i.e., rapid HIV and HCV tests) versus those that were conducted by laboratory personnel (i.e., SARS-CoV-2 antibody and RNA). 

17. Data analyses

It would be interesting to explore whether the characteristics of the participants with inconclusive SARS-CoV-2 results differ from those with positive results.

Response: We compared those testing indeterminate to the remainder of the sample and found only one significant difference between these groups. We have added the following sentences to the Results section: 

“Considering serologic evidence of SARS-CoV-2 infection, 19 (4.7%) tested indeterminate and were excluded from further analysis. These participants did not differ significantly from those who were included, with the exception that those testing indeterminate were less likely to inject heroin compared to the remainder of participants (68.4% vs. 87.4%, P=0.04).”

18. Results

I suggest some editing in this section to improve readability. It would be helpful to focus one paragraph only describing the overall sample, then another presenting the bivariate analyses, and a final one describing the multivariate analyses.

Response: We have re-ordered the Results section to improve readability as recommended, and have situated all of the descriptive statistics together. We also added subheadings to better organize the findings.

19. Discussion

This section provides information that would have been useful in the Introduction section to provide a background on the topic. This includes the prevalence and correlates of SARS-CoV-2.

Response: In response to Reviewer 1, we moved the text on prevalence and correlates of SARS-CoV-2 from the Discussion to the Introduction, and updated the references where needed (see response to #7).

20. This section presents 7 different findings obtained from the current study. It is unclear what the take home message is. I would suggest rewording and reorganizing this section to focus on the most relevant findings. 

Response: We have taken the opportunity to re-organize the beginning of the Discussion so that the opening paragraph summarizes the main findings, as follows:

“In this community-based study of people who inject drugs in the San Diego-Tijuana border region, over one third had detectable SARS-CoV-2 antibodies, which exceeds estimates from the general population in either city. This suggests that a binational policy response to COVID-19 is warranted beyond the closure of the U.S.-Mexico border. We did not observe substance use behaviors, such as smoking, vaping, or use of specific drugs such as opiates or stimulants to be associated with an elevated risk of SARS-CoV-2 infection, as others have hypothesized. Instead, we observed that circumstances in the risk environment, notably sex work and incarceration, were associated with significantly higher SARS-CoV-2 seroprevalence.” 

21. Recent or past drug use is self-reported. This should be included as a study limitation.

Response: We agree and have now specifically referred to this as a limitation.

22. Conclusions

The conclusions section does not fully mirror what is presented in the Discussion section and reads as a different paper. This last paragraph should focus on the findings related to the prevalence of SARS-CoV-2 and its correlates among PWID. The additional information is distracting.

Response: We have modified the opening paragraph of the Discussion and the Conclusions paragraph so that they are now in alignment. 

23. Minor comments: Remove “a priori” from the sentence “We hypothesized a priori that…”

Response: The editor, Dr. Page, indicated in her decision letter that this sentence should remain as originally stated.

24. Include the software used to conduct the data analyses.

Response: We added a sentence at the end of the Methods section to indicate that the analysis was conducted with SAS version 9.4.

---

## [Editor Report · Decision Letter 1]

8 Nov 2021

Prevalence and Correlates of SARS-CoV-2 Seropositivity among People who Inject Drugs in the San Diego-Tijuana Border Region

PONE-D-21-25385R1

Dear Dr. Strathdee,

We’re pleased to inform you that your manuscript has been judged scientifically suitable for publication and will be formally accepted for publication once it meets all outstanding technical requirements.

Kind regards,

Kimberly Page, PhD, MPH

Academic Editor

PLOS ONE
---

## [Editor Report · Acceptance letter]

12 Nov 2021

PONE-D-21-25385R1 

Prevalence and correlates of SARS-CoV-2 seropositivity among people who inject drugs in the San Diego-Tijuana border region 

Dear Dr. Strathdee:

I'm pleased to inform you that your manuscript has been deemed suitable for publication in PLOS ONE. Congratulations! Your manuscript is now with our production department. 

Kind regards, 

on behalf of

Dr. Kimberly Page 

Academic Editor

PLOS ONE